# Patterns of emergency admissions for ambulatory care sensitive conditions: a spatial cross-sectional analysis of observational data

Rita Santos ![ORCID],[1] Nigel Rice,[1,2] Hugh Gravelle[1]

► Prepublication history and supplemental material for this paper are available online. To view these files, please visit the journal online (http://dx.doi.org/10.1136/bmjopen-2020-039910).

[1]Centre for Health Economics, University of York, York, UK
[2]Department of Economics and Related Studies, University of York, York, UK

**Correspondence to**
Dr Rita Santos;
rita.santos@york.ac.uk

## ABSTRACT

**Objectives**  To examine the spatial and temporal patterns of English general practices' emergency admissions for Ambulatory Care Sensitive Conditions (ACSCs).

**Design**  Observational study of practice level annual hospital emergency admissions data for ACSCs for all English practices from 2004-2017.

**Participants**  All patients with an emergency admission to a National Health Service hospital in England who were registered with an English general practice.

**Main outcome measure**  Practice level age and gender indirectly standardised ratios (ISARs) for emergency admissions for ACSC.

**Results**  In 2017, 41.8% of the total variation in ISARs across practices was *between* the 207 Clinical Commissioning Groups (CCGs) (the administrative unit for general practices) and 58.2% was *across* practices within CCGs. ACSC ISARs increased by 4.7% between 2004 and 2017, while those for conditions incentivised by the Quality and Outcomes Framework (QOF) fell by 20%. Practice ISARs are persistent: practices with high rates in 2004 also had high rates in 2017. Standardising by deprivation as well as age and gender reduced the coefficient of variation of practice ISARs in 2017 by 22%.

**Conclusions**  There is persistent spatial pattern of emergency admissions for ACSC across England both within and across CCGs. We illustrate the reduction in ACSCs emergency admissions across the study period for conditions incentivised by the QOF but find that this was not accompanied by a reduction in variation in these admissions across practices. The observed spatial pattern persists when admission rates are standardised by deprivation. The persistence of spatial clusters of high emergency admissions for ACSCs within and across CCG boundaries suggests that policies to reduce potentially unwarranted variation should be targeted at practice level.

## INTRODUCTION

Ambulatory Care Sensitive Conditions (ACSCs) are conditions, such as influenza and pneumonia, diabetes, congestive heart failure, angina and chronic obstructive pulmonary disease, where good quality primary care can reduce the risk of hospital admission. Rates of emergency hospital admissions for ACSCs are used in many countries as measures of the quality of primary care and geographical variations in them as indicators of inequality.[1 2] Emergency admissions for ACSC are costly; if all local authorities (LAs) performed at the level of the best performing quintile of LAs, ACSC emergency admissions would be reduced by 18% with an associated reduction in National Health Service (NHS) expenditure of £238 million.[3]

Although there have been studies of variation across practices in rates of ACSC emergency admissions for specific conditions[4] and of trends over time in ACSC emergency admissions,[5 6] there have been no studies of the geographic variation in overall ACSC emergency admissions across general

practices. Blunt *et al*[5] show that rates of ACSC emergency admissions standardised by age, gender and deprivation were higher in 2004–2009 for Primary Care Trusts (the then administrative units for general practices) in the north of England compared with the south. NHS Right Care and Public Health England have produced maps of age and gender standardised emergency admission rates for a variety of ACSCs at Clinical Commissioning Group (CCG) level (the administrative unit to which practices belong).[7]

## Strengths and limitations of this study

► This is the first study to explore the spatial pattern of ambulatory care sensitive condition (ACSC) emergency admissions at GP practice level in England and over a substantial period of time (14 years) using indirectly standardised ACSC emergency admission ratios by age and gender and also indirectly standardised by age, gender and deprivation.

► We decompose the total variation in ACSC emergency admissions into variation between practices within administrative areas and variation across administrative areas.

► We compare changes between 2004 and 2017 in the spatial patterns of ACSC emergency admissions for conditions whose care was financially incentivised with changes in the patterns of ACSCs for conditions whose care was not incentivised.

► Understanding how much of the variation in ACSC emergency admissions is outside the influence of practices and how much is potentially amenable to policy requires patient-level data.

We make a number of contributions in this study. Since ACSC emergency admissions can be reduced by appropriate management in primary care, we examine their spatial variation at general practice level. We use spatial methods to describe the spatial pattern of practice age and gender standardised ACSC emergency admissions in England. We compare the pattern of variation at practice level with that at CCG level. We examine changes in spatial patterns of ACSC emergency admissions across practices from 2004 to 2017, both in total and for ACSCs for which care was financially incentivised via the Quality and Outcomes Framework (QOF). We test for the existence of 'hot spots' or clusters of neighbouring practices with similar unusually high (or low) ACSC admission rates that persist over time. We examine if allowing for practice level differences in deprivation, as well as age and gender, changes the spatial distribution of ACSC admission rates.

## INSTITUTIONAL BACKGROUND

The English NHS is tax-financed system and free at the point of use (apart from a small charge applied to around 10% of medicines dispensed in primary care). Most general practices are partnerships owned and run by general practitioners. On average, they have around 4 GPs, 2 nurses, 1.3 other direct patient care staff and eight administrative staff (all staff numbers are full-time equivalents) and are responsible for around 7500 patients.[8] Practices are paid by a mix of lump sum payments, capitation, quality incentive payments and items of service payments. They are reimbursed for the costs of their premises but have to fund all other expenses, such as the employment of nurses and clerical staff, from their revenue.

Practices are gatekeepers for outpatient and elective secondary care, though patients have the right to choose any qualified provider in contract with the NHS. For emergency secondary hospital care, patients self-refer or are brought in by emergency services and are almost always admitted via their nearest Accident and Emergency department.

In 2004/2005, the QOF pay for performance scheme was introduced in response to concerns over variation in quality of care provided in general practice. Practices are rewarded for achievement of indicators of clinical quality for a set of chronic conditions and process administrative quality. The QOF accounted for around 15% of practice income in 2004[9] and 8% in 2017.[10]

## DATA

Our data are generally for financial years from 1 April to 31 March. We use Hospital Episode Statistics (HES) data on all admissions between 2004 and 2017, which were coded as an emergency and admitted from a source other than a hospital ward or outpatient clinic. We use the HES patient practice code to attribute emergency admissions to practices by age and gender band (online supplemental table A1 lists data sources).

There are a variety of definitions of ACSC.[1 11–13] We use a set of ACSCs, which is the union of two partially overlapping sets proposed by the NHS Outcomes Framework[13] and Harrison *et al*.[14] In total, we use 178 ICD-10 (International Classification of Diseases, Tenth Revision) codes (online supplemental table A2) for 24 disease groups from the HES primary diagnosis field for patients with an emergency admission. This definition is broader than the used in other studies[6 15] and includes three additional disease groups: mental and behavioural disorders, cardiovascular diseases and stroke and more ICD-10 codes for some disease groups (eg, N30.0, N30.8 and N30.9 for pyelonephritis and kidney/urinary tract infections). However, our definition excludes vaccine preventable tuberculosis since emergency admissions for this condition are not classified as ACSC in NHS Outcome Framework[13] or Harrison *et al*.[14] and tuberculosis surveillance is a responsibility of Public Health England.

Management of some ACSCs was financially incentivised by the QOF, and to examine changes in these emergency admissions, we use the definition of incentivised ACSCs in Harrison *et al*.[13]

For each practice, we use NHS Digital data on the numbers of patients in 14 age and gender groups. When we standardise ACSC emergency admissions for 2017 by deprivation as well as by age and gender, we use the Attribution Data Set (ADS) (NHS Digital) and the Index of Multiple Deprivation (IMD) from ONS. ADS contains the number of practice patients resident in each Lower Super Output Area (LSOA) by age and gender band, while IMD data have an IMD score for each LSOA. From these data, we compute the number of patients in 70 age, gender and deprivation quintile groups for each GP practice.

Since very small practices may be new or in the process of merging or closing, we include practice-year observations for year $t$ only if the practice has more than 1000 patients in years $t–1$, $t$ and $t+1$. We also exclude outlier practices with more emergency admissions than patients in any age/gender band. In total, we excluded 2768 (2.5%) practice-year observations from 1928 practices. The total number of practices included in the analysis fell from 8188 in 2004 to 7340 in 2017, reflecting a trend to fewer practices with larger lists.

Practices can have more than one surgery from which they provide care. We obtained data on the location (grid reference from postcodes) of all surgeries of practices from NHS Choices and Connecting for Health archive and current data files: 17362 surgeries for 2004 and 15840 in 2017, across 8188 GP practices.

## METHODS

### Patients and Public Involvement (PPI)

A PPI group was involved in early discussions of the research topic and in discussions of the methods and presentation of results for a wider audience.

### Indirect standardisation

We calculate the Indirectly Standardised ACSC emergency Admissions Ratio (ISAR) for practice $i$ in year $t$ as

$$ISAR_{it} = \frac{Adm_{it}}{ExpAdm_{it}} 100$$

where $Adm_{it}$ is the observed number of ACSC emergency admissions in year $t$ for practice $i$, and $ExpAdm_{it}$ is the expected number of admissions. The latter is the number of admissions practice $i$ would have had in year $t$ if the age and gender group admission rates of a reference population ($RefAdmRate_g$) were applied to practice $i$'s population in those age and gender groups in year $t$:

$$ExpAdm_{it} = \sum_{g=1}^{14} RefAdmRate_g \times Pop_{igt}$$

When we examine changes in the pattern of ISARs over time (2004–2017), we compute the reference population age and gender specific admission rates as the total number of admissions in the respective groups for all practices over the full period 2004–2017. The reference population is the number of people in the practices summed across practices and years.:

$$RefAdmRate_g = \left( \sum_{t=2004}^{2017} \sum_i Adm_{igt} \right) / \sum_{t=2004}^{2017} \sum_i Pop_{igt}$$

where $Adm_{igt}$ and $Pop_{igt}$ are admissions and numbers of patients in practice $i$ in age/gender group $g$ in year $t$. This ensures that changes in practice $ISARs$ over time are only due to changes in a practice's age and gender specific admission rates, not to changes in reference admission rates or a practice's age and gender composition.

When we compare the variation in ISARs computed at practice and CCG level for 2017, we use age and gender group admission rates for 2017 to calculate expected admissions. When we standardise by deprivation, we use reference groups defined by 2017 age, gender and deprivation quintile.

### Spatial pattern analyses

#### Heat maps

We attach data on each practice's ISAR to the grid references of all of its surgeries. To depict the spatial pattern of ISARs, we impute them to all areas using Inverse Distance Weighting (IDW). This interpolation technique creates a smooth surface layer from a finite set of grid references. It is analogous to placing a light sheet over a set of spikes (grid references for surgeries) of different heights (reflecting practice ISARs). The sheet forms contours across the surface of the spikes to give a complete spatial distribution of ISARs. The ISAR imputed for a point is a weighted average of the ISARs of the 12 closest practices with weights $1/d^2$, where $d$ is the distance from the point to the nearest surgery of the practice. Thus, the mix of practice ISARs imputed for each point aims to reflect the influence of distance on patient choice of practice.[16]

### Spatial statistics

Tobler's first law of geography is that 'everything is related to everything else, but near things are more related than distant things'.[17] In the current context, this suggests that a practice's ISAR will be similar to those of nearby practices (nearest five practices): they will be spatially autocorrelated. To test if this holds, we use Moran's I statistic,[18–22] which measures the average correlation between practices ISARs in year $t$ as

$$I_t = \frac{\sum_i \sum_j \omega_{ij} \left( ISAR_{it} - I\bar{S}AR_t \right) \left( ISAR_{jt} - I\bar{S}AR_t \right)}{\sum_i \left( ISAR_{it} - I\bar{S}AR_t \right)^2},$$

where $I\bar{S}AR_t$ is the year $t$ mean of $ISAR_{it}$ over all practices and $\omega_{ij}$ is a spatial weight based on the minimum straight line distance between surgeries of practices $i$ and $j$. We set $\omega_{ij}=1$ for the five nearest practices and $\omega_{ij}=0$ otherwise. This allows the ISAR for a practice to be compared with the average ISAR of practices with overlapping catchment areas (even in rural areas) and whose patients access the same hospital trusts. Using a distance-based threshold could create very large networks for practices in urban areas and much smaller, possibly empty, networks in rural areas.

Positive values of $I_t$ indicate positive spatial autocorrelation.

Moran's I is a global spatial statistic is a measure of the extent to which the spatial pattern over all practices is randomly distributed (as opposed to spatially clustered). To find local clusters of practices with similar ISARs, we use a related indicator: Moran's Local Indicator of Spatial Association (LISA)[23]

$$I_{it} = \frac{\left( ISAR_{it} - I\bar{S}AR_t \right)}{n^{-1} \sum_j \left( ISAR_{jt} - I\bar{S}AR_t \right)^2} \sum_j \omega_{jt} \left( ISAR_{jt} - I\bar{S}AR_t \right),$$

where again we set $\omega_{ij}=1$ for the five nearest practices and $\omega_{ij}=0$ otherwise. We use the LISA statistic to identify spatial clusters of practices with similar ISARs. We denote as HH (LL) practices, which have above (below) average ISARs and are clustered within a set of nearby practices, which also have above (below) average ISARs.

## RESULTS

### Level of aggregation: CCG versus practice

Figure 1 displays the spatial pattern of ACSC ISARs in 2017 using data at two levels of aggregation. The left-hand map shows the distribution of ISARs (averaged across practices within the CCG) in each of 207 CCGs. The right-hand map has the spatial distribution for the 7340 individual practices and across 15 840 surgeries. Low (under 75) ISAR areas are shaded blue, intermediate (75–114) ISAR areas are shaded yellow and high (125 and above) are shaded red.

The maps show broadly similar spatial patterns, with higher ISARs in the North East, around Liverpool and

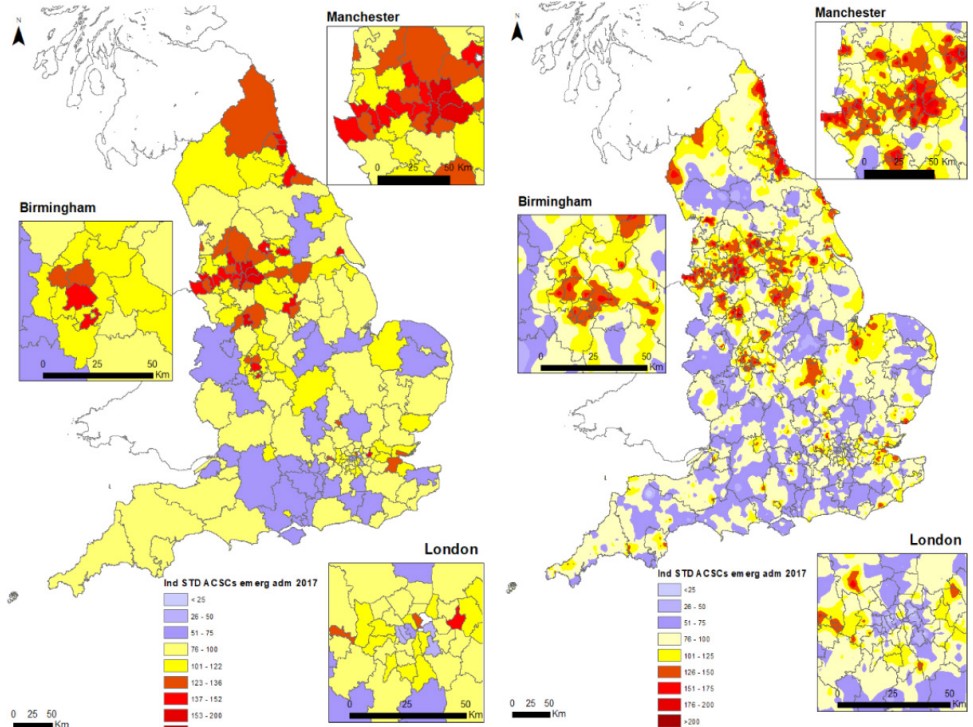

**Figure 1** CCG and practice-level ACSC emergency admission 2017. ACSC rates are indirectly standardised by age and gender with expected rates for the reference population computed from 2017 data. Grey lines are boundaries of CCGs. ACSC, ambulatory care sensitive condition; CCGs, Clinical Commissioning Groups.

Manchester, the Midlands around Birmingham and in parts of the Thames Estuary. However, a comparison across the two maps shows that CCGs with low average ISARs contain areas where practices display high levels of ISARs. We see similar heterogeneity across practices and areas for CCGs that display high levels of ISARs. For example, Northumberland CCG (in the North East) has a moderately high ISAR, but the practice level map shows that high ISARs are concentrated in seaside towns and on the border with North Tyneside CCG. Conversely, inland areas have low ISARs. There are also clusters of practices with similar ISARs that span CCG boundaries and differ from the rest of their CCGs.

The CCG maps are based on the average of their respective practice ISARs and accordingly fail to display the nuances of variation at practice level where ACSCs are managed. The coefficient of variation (SD/mean) is 0.30 at CCG level and 0.43 at practice level. More revealingly, 41.8% of the total variance in practice ISARs is between CCGs, and 58.2% is due to variation between practices within CCGs. Focusing on CCG level quality metrics is, therefore, likely to lead to an incomplete understanding of local area performance.

Our definition of ACSCs includes 24 disease groups with somewhat different spatial patterns. For example, the ISAR's spatial pattern for influenza and pneumonia is similar to that for all ACSCs, while there are a higher proportion of practices with high ISARs for Congestive Heart Failure (CHF) and stroke (online supplemental figure A1).

### Changes over time

The total number of ACSC emergency admissions increased by 28.3% between 2004/2005 and 2017/2018 (online supplemental table A3) and the unadjusted ACSC emergency admission rate increased by 11.14%. figure 2 compares the spatial pattern of age and gender adjusted ACSC ISARs for 2004 and 2017 using the same reference population (admission rates calculated across all years from 2004 to 2017) (online supplemental figure A2 maps the change between 2004 and 2017). The national mean ISAR increased from 95.12 in 2004 to 105.5 in 2013 before declining to 99.6 in 2017—an increase of 4.7% from 2004 to 2016. The increase in ISARs was not uniform. For example, in the North East high ISARs areas became more concentrated in coastal areas. Areas south of The Wash and along the Thames estuary also displayed increases in ISARs. However, in other areas, for example, the Isle of Wight, and the far South West, ACSC ISARs fell. Overall variation in ISARs, as measured by the coefficient of variation, increased from 0.378 to 0.427 over the period.

### Spatial correlations

ISARs are not randomly distributed geographically across England. Moran's global I index shows statistically significant positive spatial correlation in all years (online supplemental table A4): practice ISARs tend to be more similar to those of nearby practices than to practices further away. The LISA identifies 722 practices in 2004 with high ACSC ISARs, which were in clusters of neighbouring practices

**Figure 2** Change in spatial pattern of ACSC emergency admissions: 2004 versus 2017. ACSC rates are indirectly standardised by age and gender with expected rates for the reference population computed from data for all practices in all years 2004 to 2017. ACSC, ambulatory care sensitive condition.

that also exhibited high ACSC ratios (HH clusters) and 309 practices within spatial clusters displaying low ACSC ratios (LL clusters). The corresponding values in 2017 are 576 and 296, respectively (details in the online supplemental table A5).

Of those practices classified within an HH cluster in 2004, 70% remained in an HH cluster in 2017. Similarly, 69% of practices that were classified within an LL cluster in 2004 were also within a LL cluster in 2017 (online supplemental table A6). figure 3 shows areas that were classified as HH or LL for different lengths of time, with darker shades indicating areas belonging to clusters for longer periods.

Practices in the South and South West of England, the Midlands and the along the border with Wales exhibit the most persistent membership of LL clusters. Clusters of persistently high ACSC ratios ('hot spots') are mainly along the North East coast, Barrow-in-Furness, Liverpool, Greater Manchester, South Yorkshire and the West Midlands around Birmingham.

### Trends for ACSCs for which care was incentivised

Conditions classified as ambulatory care sensitive are those where better primary care would improve outcomes, including reducing emergency hospitalisations. The QOF was introduced in 2004 to provide financial incentives linked to indicators of care for some of these conditions. Total *unadjusted* emergency admissions for incentivised ACSCs decreased by 2.1% between 2004 and 2017. This compares to an observed increase of 28.3% for all ACSCs (online supplemental table A3).

Our comparison of trends in ISARs across time allows for changes in the size and age/gender mix of the population. There was a reduction in the year mean age and gender adjusted ISAR for incentivised conditions of 20.8% (112.52 to 89.09) from 2004 to 2017. This compares with an increase in ISAR for all ACSCs over the same period of 4.7% (95.12 to 99.6). These contrasting trends do not prove that the QOF reduced emergency admissions for incentivised ACSCs since they may just be continuations of trends that existed prior to the introduction of the QOF. However, evidence from comparison of pre-QOF and post-QOF does suggest that the QOF did reduce emergency admissions for incentivised ACSCs.[14]

Inspection of the maps in figure 4 shows that between 2004 and 2017, there were marked reductions in incentivised ACSC emergency admissions in some areas that previously displayed high ISARs, particularly in the North East and in the Liverpool-Manchester-Leeds-Hull corridor and in the South West. However, areas with initially more moderate ISARs also experienced reductions, for example in Norfolk. The overall dispersion (coefficient of variation) of incentivised ACSC ISARs increased slightly from 0.43 to 0.48 over the period of observation.

### Allowing for deprivation

Variations in practice ACSC admission rates that are due to factors outside the control of practices and CCGs are not informative for primary care policy. So far we have allowed for cross-practice variations in age and gender, but some of the cross-practice differences are due to variations in other factors not controllable by local policy, such

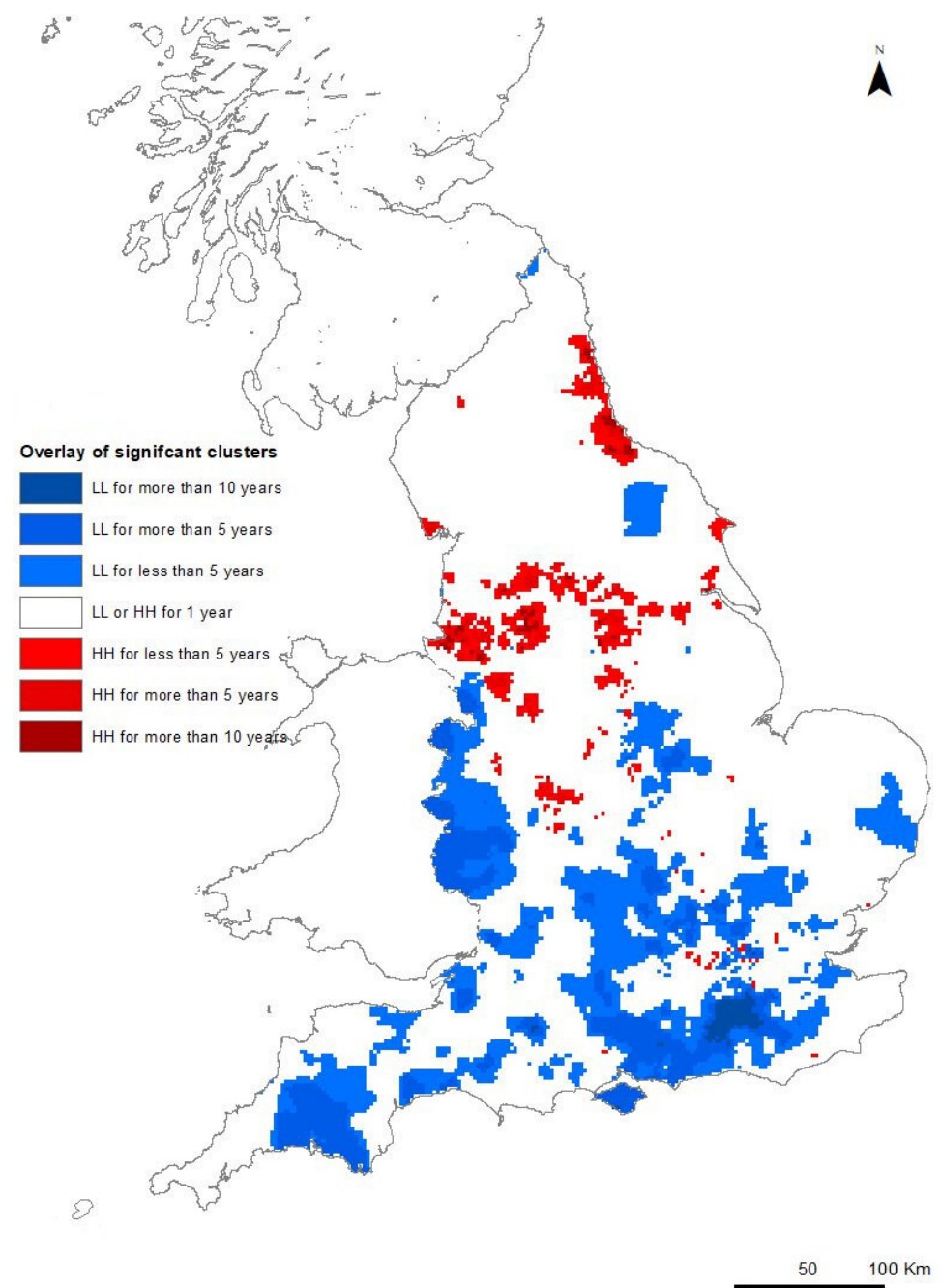

**Figure 3** Persistence of significant spatial cluster for ACSC ISARs emergency admissions from 2004 to 2017. The map shows spatial clusters, identified by local indicators of spatial association for ISARs that are statistically significant at 1%, and which persist from 2004 to 2017. HH (LL) are clusters of practices with high (low) ISARs. ACSC, ambulatory care sensitive conditions; ISARs, indirectly standardised ratios.

as deprivation.[4 14 24] Figure 5 shows the spatial pattern of ACSC ISARs after standardising by deprivation as well as by age and gender (as described in the methods section) for 2004 (left-hand panel) and 2017 (right-hand panel).

Variation is reduced after allowing for deprivation. Compared with figure 2, the maps in figure 5 that additionally allow for deprivation have more areas shaded yellow, indicating ISARs relatively close to the mean, and fewer areas shaded blue or red, indicating ISARs further from the mean. For 2017, the coefficient of variation is

reduced from 0.43 (figure 2 right-hand panel) to 0.36 (figure 5 right-hand panel). For 2004, it is reduced from 0.378 (figure 2 left-hand panel) to 0.28 (figure 5 left-hand map).

Allowing for deprivation also reduces overall clustering of practices with similar ISARs: Moran's I falls from 0.45 to 0.39 in 2017 and from 0.53 to 0.19 in 2004. The number of practices in local clusters with similar ISARs is also reduced by additionally standardising for deprivation, more so in 2004 than in 2017. In 2017, the

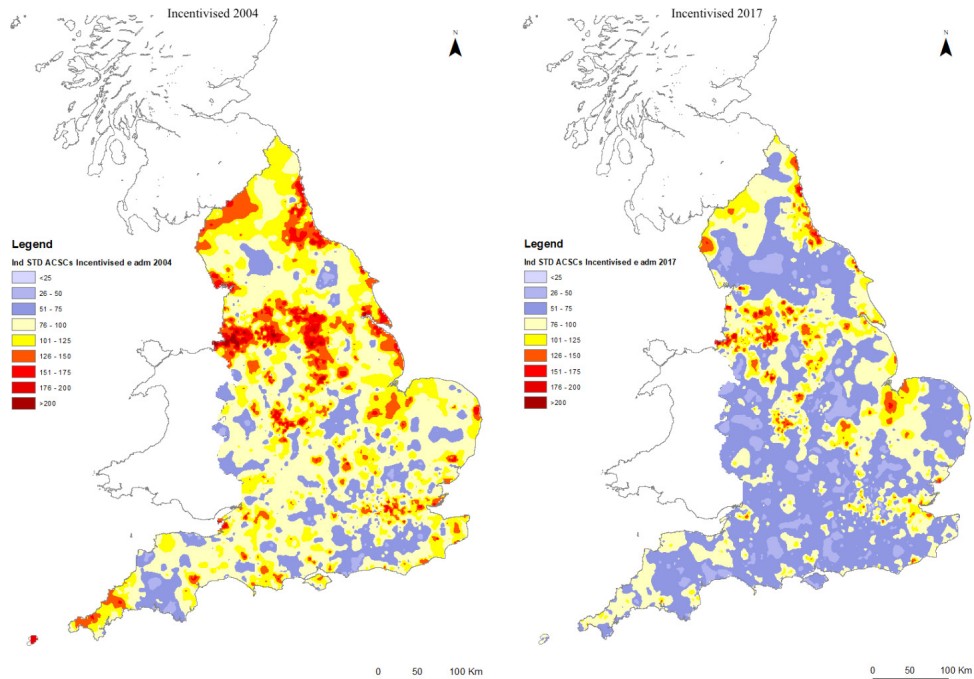

**Figure 4** ACSC for incentivised conditions 2004 and 2017. ACSC rates are indirectly standardised by age and gender with expected rates for the reference population computed from data on for all practices in all years 2004–2017 for incentivised ACSCs. ACSCs, ambulatory care sensitive conditions.

number of practices in clusters with high ISARs decrease from 576 practices (7.9%) to 228 practices (3.1%). In 2004, the corresponding values are 722 (8.8%) and 238 (3.5%). Similarly, the number in clusters with low ISARs is reduced from 296 (4.0%) to 262 to (3.6%) in 2017 and from 309 (3.8%) to 47 (0.7%).

Allowing for deprivation has different effects in different types of areas. For deprived urban coastal areas, for example, in the North East, we no longer observe high ISARs once we standardised for deprivation, whereas less deprived rural areas (eg, in the South West) display high ISARs values poststandardisation. ISARs for parts of

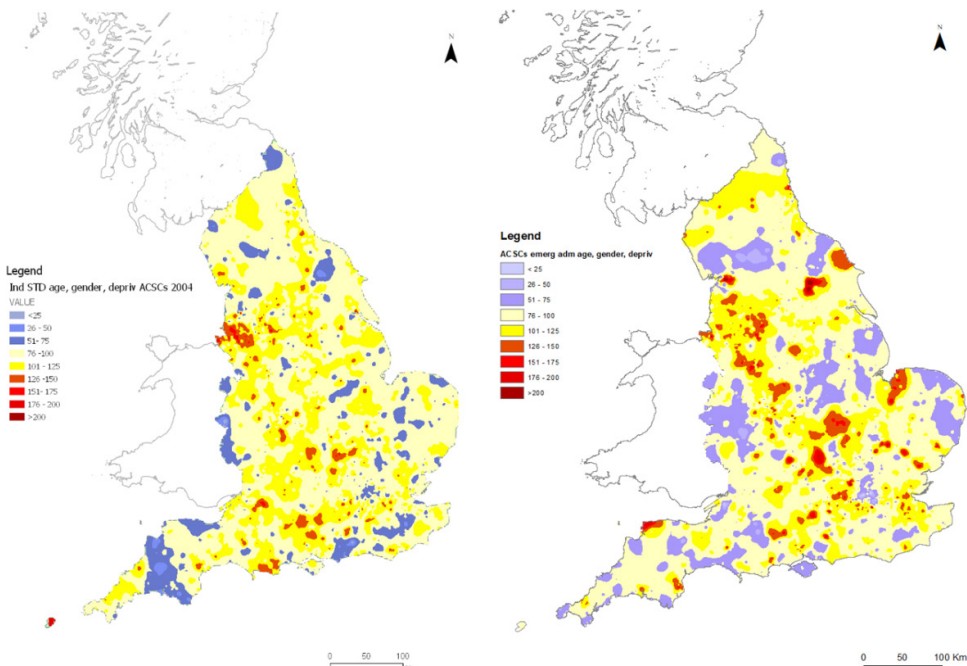

**Figure 5** Change in ACSC ISAR distribution in 2004 and 2017 after additional standardisation by deprivation. Indirect standardisation. Figures use 2004 (left-hand panel) and 2017 (light hand panel) data on admissions and practice populations to construct expected number of admissions allowing for age, gender and deprivation. ACSC, ambulatory care sensitive conditions; ISAR, indirectly standardised ratio.

Liverpool and Manchester are reduced, whereas some areas in the Midlands have higher ISARs after allowing for deprivation.

## DISCUSSION

Practice ACSC emergency admissions exhibit considerable spatial variation even after standardisation by patient age and gender. Additional standardisation by deprivation reduces this variation further, but marked differences across general practices and areas remain. There are clusters of practices with similar higher (or lower) than expected standardised ACSC admission rates. These spatial patterns persist over a considerable period of time (2004–2017). The spatial analysis also demonstrates, in line with other studies,[13] that emergency admission rates for ACSCs whose care was incentivised by the QOF fell at a faster rate than non-incentivised conditions over the study period. However, there was little change in the overall variation in emergency ACSC admissions for incentivised conditions.

Previous studies of the spatial pattern of ACSC emergency admissions have been undertaken at higher levels of spatial aggregation and have not examined trends over prolonged periods of time. Our analysis shows that mapping at the level of CCGs[6]—the administrative unit for general practice—considerably understates the full extent of variation and does not identify within CCG clusters of practices with similarly high (or low) admission rates and that often span the borders of CCGs.

We found substantial variation in an outcome of importance for primary care patients after accounting for age and gender. Additionally, standardising for deprivation, which is outside the control of practices and CCGs, but can be influenced by national policy, reduced observed variation. Allowing for deprivation had different effects in different types of areas (coastal vs inland, urban vs rural), possibly because the deprivation measure is a composite of different types of deprivation that vary across areas and that could have different effects on ACSCs.

The mapping of practice level ACSC emergency admissions standardised for age and gender is a useful method for screening for possible unwarranted variation. However, observed variation may be due to factors outside practice control. These include underlying patient morbidity and multimorbidity, coding practices and admission thresholds in local hospitals and the provision of community health and social care services by CCGs and LAs. Richer data on patients, practices (staffing, resourcing and quality), local services, the mix of hospitals used by patients and the local environment in which practices operate, combined with multivariate regression modelling, will be required to determine which practices have unduly high ACSCs emergency admissions and how much of the variation across practices is unwarranted and potentially amenable to policy intervention.

Since 1 July 2019, GP practices in England have been encouraged and funded to collaborate in Primary Care Networks (PCNs) covering populations of 30–50 000 patients.[25] In principle, this should reduce variation in outcomes, such as ACSC emergency admission, across practices within PCNs. Its possible effect on variation across PCNs, which may adopt different policies, is less obvious. The spatial methods employed in this study can be applied to examine variation within and across PCNs.

**Acknowledgements** We are grateful for comments from Tim Doran, Michael Reaker, Sheila Miller, John Gower, Kathleen Murphy, Ann Wands and Patricia Southgate and to Mark Harrison for sharing his code for extraction of ACSCs.

**Contributors** RS developed the research questions, undertook data analysis and drafted the paper. HG and NR helped develop the research questions and methods, supervised and commented on the analysis and contributed to drafting the paper. RS is the guarantor.

**Funding** This study was funded by National Institute for Health Research (grant number: NIHR Doctoral fellowship DRF 2014-07-055).

**Map disclaimer** The depiction of boundaries on the map(s) in this article does not imply the expression of any opinion whatsoever on the part of BMJ (or any member of its group) concerning the legal status of any country, territory, jurisdiction or area or of its authorities. The map(s) are provided without any warranty of any kind, either express or implied.

**Competing interests** None declared.

**Patient consent for publication** Not required.

**Provenance and peer review** Not commissioned; externally peer reviewed.

**Data availability statement** Data may be obtained from a third party and are not publicly available. Hospital Episode Statistics are copyright ©2004/05-2016/17 Health and Social Care Information Centre, DARS-NIC 84254-J2G1Q-V2.13, all rights reserved and reused with the permission of NHS Digital. No additional data available.

**ORCID iD**
Rita Santos http://orcid.org/0000-0001-7953-1960

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
