## [Reviewer comments · BMJ Open]

ARTICLE DETAILS

TITLE (PROVISIONAL)	Patterns of emergency admissions for ambulatory care sensitive conditions: a spatial cross-sectional analysis of observational data
AUTHORS	Santos, Rita; Rice, Nigel; Gravelle, Hugh

VERSION 1 – REVIEW

REVIEWER	Ben Jackson Sheffield University Faculty of Medicine, Dentistry and Health United Kingdom
REVIEW RETURNED	22-May-2020

GENERAL COMMENTS	The study focuses on the relative admission rates for ACSC over 12 years between CCGs and within CCGs at a practice level, standardised for age, gender and deprivation. All practices across England were included - it was recognised that some practices may have been created, merged or closed - and only practices that had sizeable lists (>1000) over a three year period were included. ACSC were standardised for age, gender and then deprivation, 'exposure' being standard GMS general practice. Patients will have been 'exposed' to different levels of community services outside GMS general practice. The coding accuracy of ACSC from different hospitals will have been of different quality but the authors have no control over this and it is reasonable to consider this as of standard consistency. The authors have not explicitly identified that ACSC are also managed by community health and social care services and how important these are, though by using their spacial statistics to identify practices that are close to each other and therefore will have access to similar community services, they have addressed this to some extent. I think a recognition of the significance of care outside GMS general practice should be a minor revision - particular in the context of publication in an international journal. In their analysis of IMD they have described how the additional analysis by deprivation altered rural and coastal areas more than others - but have not suggested any possible reasons for this. It could be that the relatively high component score of availability to services over more socio-economic factors within the overall IMD score may be a factor. The findings are important and in line with other literature. Variation is at practice level as well as CCG level. IMD does not reduce variation but reduces it - presumably through reducing the relative impact of primary care services against overall IMD factors - this is an interesting paper and opens up other areas of research into
---

	what additional factors outside the control of practices might be contributing to the variation. The paper has relevance and provides support for the policy of focusing on PCNs outcomes rather than CCGs, and the importance of addressing variance within practices - it does not provide evidence of the best way of addressing this variance. The fact that this variation persists despite IMD suggests that there are additional service factors at play and these should be addressed for those communities.
--	--

REVIEWER	Laia Maynou Health Policy, London School of Economics and Political Science, UK
REVIEW RETURNED	20-Jul-2020

GENERAL COMMENTS	bmjopen-2019- 039910: Patterns of emergency admissions for ambulatory care sensitive conditions: a spatial analysis of observational data The aim of this paper is to examine changes in spatial patterns of ACSC admissions across general practices from 2004 to 2017. While it is very interesting and relevant for the NHS, it would benefit from a revision: 1) In the introduction, it would be good to give a more detailed description of ACSC – give a list of the conditions (for instance, just the once with the higher admissions), e.g. Influenza and pneumonia, COPD, congestive heart failure, angina, diabetes... The coding used in this paper for the ACSC is a combination of NHS Outcomes Framework and Harrison et al. (2013). However, there is a more recent paper, Wallace et al (2016) that also provides an ACSC coding taking into account primary and secondary diagnoses codes. Wallace, E., Smith, S., Fahey, T. & Roland, M. (2016), 'Reducing emergency admissions through community based interventions', BMJ352, h6817. 1) The contribution of the paper is vaguely explained. More details are needed in the introduction. 2) While your interest is at the GP level, the admissions are at the hospital level. In the institutional background, it would be good to also see the pathway from GP to emergency admission. GPs are the gatekeepers but for these particular conditions (which should be treated at primary care level), they end up being emergency admissions. 3) While it is interesting to look at the total ACSC, it would have been interesting to also disaggregate it by conditions – at least, for the ones with a higher proportion. Is there any spatial variation across conditions? This type of analysis could identify some variation in practice across areas. 4) For the spatial analysis, you take into account the 5 closest practices. Is that a standard option for this method? Wouldn't it be better to do it by distance? For instance, take into account the
---

	number of practices within a radius distance of 30km (standard measure in hospital competition - (Longo et al., 2017; Gutacker 2016; Bloom et al., 2015; Gaynor, Laudicella, & Propper, 2012). I know this is at the GP level, so, 30km might seem a lot, but using 5 practices could be quite arbitrary... Is there a way to justify it? 5) Very interesting descriptive analysis at the practice and CCG level. I would have said to do it at the trust level, but that should be quite similar to CCGs...The temporal analysis – persistence – is also interesting because it could have changed over time. Both analyses are correctly explained, but the analysis that looks at the incentivised conditions is not really explained until you get to the results. It would be good to explain it in the data or methods section. 6) When presenting the maps were ISAR is adjusted by deprivation too, it would be interesting to repeat the temporal analysis for this new measure. Because these results are only based on 2017. 7) In the discussion, it would be interesting to explain the policy implications of these results. Should the NHS pay more attention to the areas with higher ACSC admissions? Are the incentives useful to reduce them? You have a couple of points on that but a bit more details would be interesting. Is this paper in line with previous literature?
--	---

VERSION 1 – AUTHOR RESPONSE

Reviewer 1

Reviewer Name: Ben Jackson

Thank you for your helpful comments on our paper. We have reproduced them below in italics with our responses in plain text.

1. The study focuses on the relative admission rates for ACSC over 12 years between CCGs and within CCGs at a practice level, standardised for age, gender and deprivation.

All practices across England were included - it was recognised that some practices may have been created, merged or closed - and only practices that had sizeable lists (>1000) over a three year period were included.

ACSC were standardised for age, gender and then deprivation, 'exposure' being standard GMS general practice. Patients will have been 'exposed' to different levels of community services outside GMS general practice.... The coding accuracy of ACSC from different hospitals will have been of different quality but the authors have no control over this and it is reasonable to consider this as of standard consistency.

The authors have not explicitly identified that ACSC are also managed by community health and social care services and how important these are, though by using their spatial statistics to identify practices that are close to each other and therefore will have access to similar community services, they have addressed this to some extent.

I think a recognition of the significance of care outside GMS general practice should be a minor revision - particular in the context of publication in an international journal.

We now mention in the Discussion section the possibility that variation could also be due to the effects of other services on ACSCs or local hospital coding practices and admission

thresholds and that this should be allowed for in assessing the extent of unwarranted variation at practice level.

Page 14: “The mapping of practice level ACSC emergency admissions standardised for age and gender is a useful method for screening for possible unwarranted variation. But observed variation may be due to factors outside practice control. These include underlying patient morbidity and multi-morbidity, coding practices and admission thresholds in local hospitals, and the provision of community health and social care services by CCGs and local authorities. Richer data on patients, practices (staffing, resourcing, and quality), local services, the mix of hospitals used by patients, and the local environment in which practices operate, combined with multivariate regression modelling, will be required to determine how much of the variation across practices is unwarranted and potentially amenable to policy intervention.”

In their analysis of IMD they have described how the additional analysis by deprivation altered rural and coastal areas more than others - but have not suggested any possible reasons for this. It could be that the relatively high component score of availability to services over more socio-economic factors within the overall IMD score may be a factor.

The findings are important and in line with other literature. Variation is at practice level as well as CCG level. IMD does not reduce variation but reduces it - presumably through reducing the relative impact of primary care services against overall IMD factors - this is an interesting paper and opens up other areas of research into what additional factors outside the control of practices might be contributing to the variation.

As the reviewer highlights, the index of multiple deprivation includes a health deprivation and disability domain. This domain includes, in 2004 and 2015, variables: Years of Potential Life Lost, Comparative Illness and Disability Ratio, an age and gender standardised rate of emergency admissions to hospital, and Adults under 60 suffering from mood or anxiety disorders. The domain has a weight of 13.5% for the IMD in 2004 and 2015.

We now note in the Discussion that inclusion of different domains of deprivation in the overall deprivation measure could account for the different effects on allowing for deprivation across different types of areas:

Page 14. “...Allowing for deprivation had different effects in different types of areas (coastal versus inland, urban versus rural), possibly because the deprivation measure is a composite of different types of deprivation which vary across areas and which could have different effects on ACSCs.”

The paper has relevance and provides support for the policy of focusing on PCNs outcomes rather than CCGs, and the importance of addressing variance within practices - it does not provide evidence of the best way of addressing this variance. The fact that this variation persists despite IMD suggests that there are additional service factors at play and these should be addressed for those communities.

We now note the introduction of PCNs in the Discussion section and explain that there is need for local coordination of services to address the persistent high ISAR spatial clusters.

Page 14: “...Since 1st July 2019, GP practices in England have been encouraged and funded to collaborate in Primary Care Networks (PCNs) covering populations of 30–50,000 patients.²⁴ In principle this should reduce variation in outcomes, such as ACSC emergency admission, across practices within PCNs. Its possible effect on variation across PCNs which may adopt different policies is less obvious. The spatial methods employed in this study can be applied to examine variation within and across PCNs.”

Reviewer 2:

Reviewer Name: Laia Maynou

Thank you for your very helpful comments on our paper. We have sought to address all the issues raised together with those of the other referee and editor.

Our responses to your specific points are detailed below in standard text. Your comments are italicised.

The aim of this paper is to examine changes in spatial patterns of ACSC admissions across general practices from 2004 to 2017. While it is very interesting and relevant for the NHS, it would benefit from a revision:

1) In the introduction, it would be good to give a more detailed description of ACSC – give a list of the conditions (for instance, just the once with the higher admissions), e.g. Influenza and pneumonia, COPD, congestive heart failure, angina, diabetes...

We now give some examples of high volume ACSCs in the Introduction:

Page 4: “ Ambulatory Care Sensitive Conditions (ACSC) are conditions, such as influenza and pneumonia, diabetes, congestive heart failure, angina, chronic obstructive pulmonary disease, where good quality primary care can reduce the risk of hospital admission.”

The coding used in this paper for the ACSC is a combination of NHS Outcomes Framework and Harrison et al. (2013). However, there is a more recent paper, Wallace et al (2016) that also provides an ACSC coding taking into account primary and secondary diagnoses codes.

Wallace, E., Smith, S., Fahey, T. & Roland, M. (2016), ‘Reducing emergency admissions through community based interventions’, BMJ352, h6817.

Thank you for raising this point. Wallace et al (2016) uses the same ACSCs emergency admissions as Bardsley et al (2013), which is originally from the State Health Department of Victoria, Australia. Our ACSC definition is broader than that used by these studies since it also includes mental and behavioural disorders, cardiovascular diseases, and stroke and has more ICD 10 codes for some disease groups. We exclude vaccine preventable tuberculosis as it is less likely to be linked to GP practice quality and tuberculosis surveillance is part of the remit of Public Health England. (https://assets.publishing.service.gov.uk/government/uploads/system/uploads/attachment_data/file/520455/PHOF_cons_response.pdf).

Bardsley et al (2013) also include secondary diagnosis for a set of conditions (gangrene, diabetes complications, pneumonia and other vaccine preventable conditions) which were responsible for a threefold increase and accounted for most of the additional growth in ACSC over non-ACSC admissions.

We now explain this in the Data section, page 6.

“...This definition is broader than the used in other studies^{5, 14}, and includes three additional disease groups; mental and behavioural disorders, cardiovascular diseases and stroke, and more ICD 10 codes for some disease groups (for example, N30.0, N30.8 and N30.9 for pyelonephritis and kidney/urinary tract infections). However, our definition excludes vaccine preventable tuberculosis since emergency admissions for this condition are not classified as ACSC in NHS Outcome Framework¹² or Harrison et al.¹³ and tuberculosis surveillance is a responsibility of Public Health England. We also use this[N1][RS2] definition to analyse the set of QOF incentivised ACSCs.”

1) The contribution of the paper is vaguely explained. More details are needed in the introduction.

We have altered the text in the introduction to better explain the contribution of the paper. Please see page 4.

Page 4: “We make a number of contributions in this study. Since ACSC emergency admissions can be reduced by appropriate management in primary care we examine their spatial variation at general practice level. We use spatially modelling methods to describe the spatial pattern of practice age and gender standardised ACSC emergency admissions in England. We compare the pattern of variation at practice level with that at CCG level. We examine changes in spatial patterns of ACSC admissions across practices from 2004 to 2017, both in total and for ACSCs for which care was financially incentivised. We test for the existence of ‘hot spots’ or clusters of neighbouring practices with similar unusually high (or low) ACSC admission rates which persist over time. We examine if allowing for practice level differences in deprivation, as well as age and gender, changes the spatial distribution of ACSC admission rates.”

2) While your interest is at the GP level, the admissions are at the hospital level. In the institutional background, it would be good to also see the pathway from GP to emergency admission. GPs are the gatekeepers but for these particular conditions (which should be treated at primary care level), they end up being emergency admissions.

Thanks you for pointing this out - we have added an explanation of the pathway from HP to emergency admission in the section on institutional background (page 5).

“Practices are gatekeepers for outpatient and elective secondary care, though patients have the right to choose any qualified provider in contract with the NHS. For emergency secondary hospital care, patients self refer or are brought in by emergency services, and are almost always admitted via their nearest Accident and Emergency Department (AED).”

3) While it is interesting to look at the total ACSC, it would have been interesting to also disaggregate it by conditions – at least, for the ones with a higher proportion. Is there any spatial variation across conditions? This type of analysis could identify some variation in practice across areas.

We have now included maps of the ISAR spatial pattern for three of the disease group with the highest number of emergency admissions. The maps are displayed in Appendix Figure 1A. and mentioned on page 10:

“Our definition of ACSCs includes 24 disease groups with somewhat different spatial patterns. For example, the ISAR’s spatial pattern for flu and pneumonia is similar to that for all ACSCs, while there are a higher proportion of practices with high ISARs for CHF and Stroke (Supplementary Figure A1).”

4) For the spatial analysis, you take into account the 5 closest practices. Is that a standard option for this method? Wouldn’t it be better to do it by distance? For instance, take into account the number of practices within a radius distance of 30km (standard measure in hospital competition - (Longo et al., 2017; Gutacker 2016; Bloom et al., 2015; Gaynor, Laudicella, & Propper, 2012). I know this is at the GP level, so, 30km might seem a lot, but using 5 practices could be quite arbitrary... Is there a way to justify it?

In Santos et al (2016) we found that in an urban area there could be more than 30 GP practices within 10km, whereas in a rural areas there may be a single practice within the same distance. Most patients are registered with a practice within 2km of their LSOA centroid.

We explain our rationale for selecting the five nearest neighbours spatial weight matrix in the methods section, page 8 and 9.

“We set $\omega_{ij} = 1$ for the five nearest practices and $\omega_{ij} = 0$ otherwise. This allows the ISAR for a practice to be compared with the average ISAR of practices with overlapping catchment areas and whose patients access the same hospital trusts. Using a distance based threshold could create very large networks for practices in urban areas and much smaller, possibly empty, networks in rural areas.”

Santos R, Gravelle H, Propper C. (2017) “Does quality affect patients’ choice of doctor? Evidence from the UK.” *The Economic Journal*, vol 127, issue 600, pp. 445-494, DOI: 10.1111/econj.12282.

5) Very interesting descriptive analysis at the practice and CCG level. I would have said to do it at the trust level, but that should be quite similar to CCGs...The temporal analysis – persistence – is also interesting because it could have changed over time. Both analyses are correctly explained, but the analysis that looks at the incentivised conditions is not really explained until you get to the results. It would be good to explain it in the data or methods section.

We have included additional text to mention the incentivised conditions analysis earlier in the paper. The introduction (page 4) now reads: “We examine changes in spatial patterns of ACSC admissions across practices from 2004 to 2017, both in total and for ACSCs for which care was financially incentivised via the QOF.”

In the data section (page 6) we have added: “Management of some ACSCs was financially incentivised by the QOF and to examine changes in these emergency admissions we use the definition of incentivised ACSCs in Harrison et al.¹³”

6) When presenting the maps where ISAR is adjusted by deprivation too, it would be interesting to repeat the temporal analysis for this new measure. Because these results are only based on 2017.

We have now included the ISAR standardised by age, gender and deprivation for 2004 in Figure 5, and discuss the changes in observed variation in the test. The coefficient of variation increases from 0.28 to 0.36 from 2004 to 2017.

We have amended the relevant discussion in Results section, page 12:

“Figure 5 shows the spatial pattern of ACSC ISARs after standardising by deprivation as well as by age and gender (as described in the methods section) for 2004 (left hand panel) and 2017 (right hand panel).

Variation is reduced after allowing for deprivation. Compared with Figure 2, the maps in Figure 5 which additionally allow for deprivation have more areas shaded yellow, indicating ISARs relatively close to the mean, and fewer areas shaded blue or red, indicating ISARs further from the mean. For 2017 the coefficient of variation is reduced from 0.43 (Figure 2 right hand panel) to 0.36 (Figure 5 right hand panel). For 2004 it is reduced from 0.378 (Figure 2 left hand panel) to 0.28 (Figure 5 left hand map).

Allowing for deprivation also reduces overall clustering of practices with similar ISARs: Moran’s I falls from 0.45 to 0.39 in 2017 and from 0.53 to 0.19 in 2004. The number of practices in local clusters with similar ISARs is also reduced by additionally standardising for deprivation, more so in 2004 than

in 2017. In 2017 the number of practices in clusters with high ISARs decrease from 576 practices (7.9%) to 228 practices (3.1%). In 2004 the corresponding values are 722 (8.8%) and 238 (3.5%). Similarly, the number in clusters with low ISARs is reduced from 296 (4.0%) to 262 to (3.6%) in 2017 and from 309 (3.8%) to 47 (0.7%).

7) In the discussion, it would be interesting to explain the policy implications of these results. Should the NHS pay more attention to the areas with higher ACSC admissions? Are the incentives useful to reduce them? You have a couple of points on that but a bit more details would be interesting. Is this paper in line with previous literature?

We now note in the Discussion that our results on incentivised conditions are in line with those in Harrison et al. However, we caution against using this type of analysis to choose areas where policy should be directed. Rather we think it is a useful screening device for the extent of, and changes in, unwarranted variation, to identify areas where further investigation might be required. We suggest that this type of spatial analysis could be useful, for example, in investigating the impact of PCNs.

“Discussion

Practice ACSC emergency admissions exhibit considerable spatial variation even after standardisation by patient age and gender. Additional standardisation by deprivation reduces this variation further but marked differences across general practices and areas remain. There are clusters of practices with similar higher (or lower) than expected standardised ACSC admission rates. These spatial patterns persist over a considerable period of time (2004-2017). The spatial analysis also demonstrates, in line with other studies,¹³ that emergency admission rates for ACSCs whose care was incentivised by the Quality and Outcomes Framework fell at a faster rate than non-incentivised conditions over the study period. However, there was little change in the overall variation in emergency ACSC admissions for incentivised conditions.

Previous studies of the spatial pattern of ACSC emergency admissions have been undertaken at higher levels of spatial aggregation and have not examined trends over prolonged periods of time. Our analysis shows that mapping at the level of Clinical Commissioning Groups⁶ – the administrative unit for general practice – considerably understates the full extent of variation and does not identify within CCG clusters of practices with similarly high (or low) admission rates and which often span the borders of CCGs.

We found substantial variation in an outcome of importance for primary care patients after accounting for age and gender. Additionally, standardising for deprivation, which is outside the control of practices and CCGs, but can be influenced by national policy, reduced observed variation. Allowing for deprivation had different effects in different types of areas (coastal versus inland, urban versus rural), possibly because the deprivation measure is a composite of different types of deprivation which vary across areas and which could have different effects on ACSCs.

The mapping of practice level ACSC emergency admissions standardised for age and gender is a useful method for screening for possible unwarranted variation. But observed variation may be due to factors outside practice control. These include underlying patient morbidity and multi-morbidity, coding practices and admission thresholds in local hospitals, and the provision of community health and social care services by CCGs and local authorities. Richer data on patients, practices (staffing, resourcing, and quality), local services, the mix of hospitals used by patients, and the local environment in which practices operate, combined with multivariate regression modelling, will be required to determine which practices have unduly high ACSCs emergency admissions and how much of the variation across practices is unwarranted and potentially amenable to policy intervention. Since 1st July 2019, GP practices in England have been encouraged and funded to collaborate in Primary Care Networks (PCNs) covering populations of 30–50,000 patients.²⁴ In principle this should reduce variation in outcomes, such as ACSC emergency admission, across practices within PCNs. Its possible effect on variation across PCNs which may adopt different policies is less obvious.

The spatial methods employed in this study can be applied to examine variation within and across PCNs.”

[N1]I've changed this from “the latter” to this as I'm assuming you mean the same definition as discussed above rather than something else.

[RS2]Yes, we use Harrison et al.

VERSION 2 – REVIEW

REVIEWER	Laia Maynou LSE, UK
REVIEW RETURNED	22-Sep-2020

GENERAL COMMENTS	The authors have done a great job answering my comments. I would recommend the manuscript for publication.
--